# A Localization Method of Ant Colony Optimization in Nonuniform Space

**DOI:** 10.3390/s22197389

**Published:** 2022-09-28

**Authors:** Qin Xu, Lei Zhang, Wenjuan Yu

**Affiliations:** 1Institution of Data Science, City University of Macau, Macau 999078, China; 2Department of Traffic Information and Control Engineering, Tongji University, Shanghai 200070, China

**Keywords:** ant colony optimization, spatial modeling, quadtree, diffusion mechanism, site location

## Abstract

The purpose of geographic location selection is to make the best use of space. Geographic location selection contains a large amount of spatiotemporal data and constraints, resulting in too many solutions. Therefore, this paper adopts the ant colony algorithm in the meta-heuristic search method combined with the incomplete quadtree to improve the searchability of the space. This paper proposes an improved ant colony algorithm in nonuniform space to solve the P-center facility location problem. The geographic space is divided by the incomplete quadtree, and the ant colony path is constructed on the level of the quadtree division. Ant colonies can leave pheromones on multiple search paths, and optimized quadtree encoding in nonuniform space stores pheromone matrices and distance matrices. The algorithm proposed in this paper improves the pheromone diffusion algorithm and the optimization objective at the same time to update the pheromone in the nonuniform space and obtain the ideal solution. The results show that the algorithm has excellent performance in solving the location problem with good convergence accuracy and calculation time.

## 1. Introduction

With the development of the economy, people are concentrated in cities. In recent years, the problem of contradictions between people and land has increasingly become a hot spot of society. The occurrence of various types of contradictions between people and land cannot be attributed to the growth of urban populations. On the contrary, scientific spatial planning and facility location can greatly alleviate the problem of contradictions between people and land. In contemporary cities, single regional planning is unable to meet the needs of people’s life, such as roads, public facilities, transportation hubs, etc. There are structural problems such as uneven spatial distribution and mismatch between geographical location and economic benefits. Therefore, for most target facilities, it is necessary to plan multiple areas for site selection. However, for the location selection of various target facilities, their coverage, path cost between areas and the importance of the location of facilities to nearby areas are considered. Basing site selection on the above factors will help cities meet the daily needs of urban residents with less expenditure. Therefore, location selection to make spatial planning, management and utilization have the best effectiveness can alleviate the increasingly prominent contradiction between people and land.

Hakimi was the first to systematically expound the problem of facility location in terms of development order and theoretical importance, and he proposed the problem of the number of P-centers [1,2,3]. In addition, typical problems in site selection research, such as the P-median [4,5] problem and the coverage problem [6] are common problems in current site selection research. Later, researchers proposed different classical optimization algorithms for facility location problems from various perspectives to meet practical needs. The solution strategy of the classical optimization algorithm generally depends on the objective type, constraints and variable types. The effectiveness of classical optimization algorithms depends on the solution space, the number of decision variables and the number of constraints in the modeling. Therefore, the solution to the P-center problem in this period is constrained to a specific situation and a small scale. However, as the P-center problem is gradually integrated into the real world, various deformations and extensions begin to appear. In most real-world problems, the solution space of the P-center problem is infinite or too large for all solutions to be evaluated. In most cases, this requires some assumptions, but the validity of these assumptions can be difficult to accept. The above are the shortcomings of classical optimization algorithms in dealing with large-scale combinatorial problems and nonlinear problems [7].

At the same time, the use of meta-heuristic algorithms with evolutionary characteristics to solve the problem of facility location has become a research trend in spatial search optimization. general-purpose meta-heuristic methods are evaluated in nine different groups: biology-based, physics-based, social-based, music-based, chemical-based, sport-based, mathematics-based, swarm-based and hybrid methods which are combinations of these [8]. Genetic algorithm [9], ant colony algorithm [10,11] and tabu search algorithm [12] are biologically based; simulated annealing algorithm [13,14] are physics-based [15,16]. A meta-heuristic is an algorithm based on intuition or empirical construction that gives a feasible solution to a problem at an acceptable cost (referring to computing time and space). Compared with the P-center problem, which can only be solved under ideal conditions, meta-heuristic algorithms are gradually applied to larger-scale and more complex real-world problems. However, they still have the following problems. First, the problem of facility location in space is an NP-hard problem [1,17,18]. As the scale of spatial data increases, the temporal and spatial complexity increase exponentially, and optimization at the spatial search algorithm level cannot solve the problem of computational scale. Second, in urban life, the state of some areas is relatively stable for a period of time and has a static nature, while in most cases the state of the area changes with time, seasons and other factors, such as population flow in urban areas changing densely populated areas and sparse areas. Therefore, comprehensive consideration of the static and dynamic properties of urban areas is also one of the complex situations faced by the location selection of target facilities [19,20,21]. Third, in the real urban space, the size of the facility area that needs to be planned is not uniform, and there are also unusable areas of different sizes and shapes in the geographic space of the city [22,23]. Therefore, for the modeling problem of urban geographic space, while taking into account the spatial accuracy and complexity, scholars generally use raster-based two-dimensional spatial modeling. This is a simple but efficient way to model space, and the biggest advantage of raster space is that it is easy to combine with heuristics. The raster method consists of a two-dimensional raster and the information of each raster. The resolution of the raster is an important parameter that directly affects the algorithm’s ability to find paths in a dense environment, the amount of environmental information storage and the computing time. However, the limitation of the raster method is that the raster size is fixed and difficult to change, and it wastes a lot of computing resources for unavailable areas in urban geographic space that cannot effectively meet the dynamic changes of urban areas.

In order to solve the problem of facility location in urban space, this paper optimizes from two aspects of spatial modeling and search algorithms. First, this paper uses the idea of incomplete quadtree-based nonuniform space modeling [24]. The purpose is to solve the problem that the modeling results of the raster method can only run the spatial search algorithm with the inherent accuracy, resulting in huge time and space costs. In nonuniform spaces such as cities, recursion is used to divide sparse, dense and unavailable areas into grids, and abstract these into the split and merge of quadtree nodes to finally generate incomplete quadtrees [25]. This enables the subsequent search algorithm to perform hierarchical searches and find solutions in the geographic space and to speed up the processing speed for the spatial data of large areas. The data structure that uses the incomplete quadtree to store hierarchical information is suitable for fast data processing and has a wide range of applications in other fields such as database and image processing. Secondly, the ant colony algorithm is used in the space search algorithm. Different from the ant colony algorithm in the traveling salesman problem, the ant colony algorithm based on quadtree search needs to optimize the encoding and storage methods of the pheromone matrix and the distance matrix [26]. At the same time, in order to consider the coverage of the facility area, the path cost between each area and the importance of facility location to the nearby area, the ant colony algorithm is improved by combining pheromone diffusion and adopting different pheromone calculation methods according to the optimization target [27].

The modeling method of a multi-way tree in nonuniform space enables hierarchical division of geographic space. The ant colony algorithm uses the root node as the starting point to locate the raster cell on the generated quadtree. We then take advantage of the quadtree hierarchical information search method to construct a search path. Ants gradually leave pheromones on the search path of the quadtree and use the communication of pheromones to indirectly cooperate to obtain ideal candidate solutions to solve the problem of spatial location configuration or location selection. The first part of this paper introduces the motivations behind the ant colony algorithm in nonuniform space. The second part introduces the modeling process in nonuniform space and the operation process of the ant colony algorithm in nonuniform space, including the diffusion of pheromone in nonuniform space and the update of the optimization objective. The final, third part is the experimental setting, results and conclusion of the article.

## 2. Motivations

City Positioning System: The modern urban positioning system is divided into four layers based on basic urban resources: perception layer, communication layer, data layer and application layer [28]. The perception layer is the primary link of the city positioning technology system. It mainly uses Global Position System (GPS), Beidou System (BDS) and other sensing equipment and technologies to collect and process location and status information. It realizes the comprehensive perception of people and objects in the city. High-precision perception layer equipment provides the basis for the efficient operation of the city. The collection results of the perception layer are connected to the communication layer through various perception control networks. The communication layer provides safe, reliable and timely data transmission and realizes various types of data fusion. Both the perception layer and the communication layer are infrastructure services for urban construction. Above the communication layer is the data layer, which integrates the original data obtained by the perception layer into the corresponding professional database according to different domain models and establishes the urban information warehouse according to the time and space dimensions. The positioning method of ant colony intelligence in nonuniform space proposed in this paper mainly relies on geographic information system (GIS) data to realize urban spatial modeling and efficient utilization of urban areas in the data layer. We expect to integrate multiple relatively low-cost computing entities, including spatial modeling and positioning algorithms, into a system with high precision and powerful computing power. Finally, the application layer directly faces the end-users of the city and provides diversified applications and services.

Dynamic Deployment in Urban Areas: Hot-spots in urban life change with time, and the spatial states at different times are also different, so repeating spatial modeling for cities wastes a lot of resources. In order to solve this problem. The nonuniform spatial modeling mentioned in this paper can effectively reduce the complexity of urban modeling. At the same time, combined with the pheromone update method in the ant colony algorithm, the nonuniform spatial modeling mentioned in this paper can dynamically update the model and regional location positioning according to the urban state at different time periods. This method reduces repetitive modeling operations. The algorithm proposed in this paper can be applied to the location selection of emergency facilities, and the location selection is more extensive, such as the location selection of public services such as police stations, fire stations and hospitals. Further, the algorithm proposed in this paper can be applied to the changing logistics, traffic and police deployment situations.

## 3. Nonuniform Space Modeling

In the two-dimensional space, a linear quadtree is used to recursively divide the nonuniform space, and the division termination conditions are set according to various complex states in the space. The entire two-dimensional space is divided into multiple raster cells, and the main idea of linear quadtree division space is as follows:

### 3.1. Obtain the Two-Dimensional Space of the City

Obtain the two-dimensional space of the city and set the total length and width of the space as (H,W). The weight is used to represent the importance of regional comprehensive factors, with a weight equal to 0 indicating that it is non-selectable or has a low degree of importance. Generate a two-dimensional weight matrix according to the urban space, assuming that the shape of the acquired space is a rectangle. If the selected space is irregular, you can set the weight of the non-selectable region in the matrix to 0. There are also some regions within the city that are not options. Therefore, the non-selectable terrain in the three-dimensional can be projected on two-dimensional space, and the weight of the obstacle area generated by the projection can be set to 0.

### 3.2. Recursively Divide Space

Set the length and width of the raster to (h,w). The region uses the quadtree method to recursively divide the two-dimensional space. It is assumed to be an ideal state, that is, the length and width of the raster cells in the uniform space are equal to (h,w) to stop the division. Ideally, the total number of cells is $4^{layer-1}$.

The formula for the length and width of a raster cell is as follows: (1)$$h=\frac{H}{2^{layer-1}}$$
(2)$$w=\frac{W}{2^{layer-1}}$$
where layer is the depth of the quadtree, which is equal to the number of layers divided into the raster space.

However, in the nonuniform space, a weight threshold can be set. If the raster weight is lower than the threshold or the raster length and width are equal to (h,w), the division will be stopped. Therefore, let the total number of rasters in nonuniform space be *N*: (3)$$N\leq 4^{layer-1}$$

When the total number of rasters *N* is less than $4^{layer-1}$, it proves that there are rasters of nonuniform size, which means that there are some lengths and widths of raster: (4)$$h>\frac{H}{2^{layer-1}}$$
(5)$$w>\frac{W}{2^{layer-1}}$$

The quadtree is generated based on the division process of the raster space. The entire raster space is regarded as the root node, and the value of the root node is equal to the sum of the weights of all raster units. Each raster unit is used as a leaf node, and the weight of the ancestor node of each leaf node is the accumulation of the weights of the included leaf nodes, thereby generating a quadtree.

## 4. The Positioning Method of Ant Colony Intelligence

Based on the quadtree structure constructed by the above two-dimensional space division, the ant colony localization algorithm is used to select the location of multi-location regions on the basis of the divided raster units. Before running the ant colony algorithm, it is necessary to initialize the pheromone matrix and the distance matrix according to the quadtree node. Taking the root node as the starting point of the ant colony, the transition probability of each optional node is calculated until it reaches the leaf node, which is used as the location region. The pheromone of the selected node is updated after each generation of ant colony is completed.

### 4.1. Pheromone Matrix

In Figure 1, each quadtree in the matrix has a corresponding number. In the raster space divided by the quadtree, the upper left is 0, the upper right is 1, the lower left is 2, and the lower right is 3.

The pheromone matrix needs to be initialized before running the ant colony localization algorithm on the quadtree to select the target facility. Figure 2 shows the initialization of the pheromone matrix and assignment of the weights of each node in the quadtree to the corresponding elements in the pheromone matrix. The dimension of the pheromone matrix is $\left[layer,4^{layer-1}\right]$; the layer is the depth of the quadtree, and the nodes that do not exist are set to 0 in the pheromone matrix. Figure 2 shows the process of initializing the pheromone matrix of the three-layer quadtree. It can be observed that the shape of the pheromone matrix is similar to the lower triangular matrix.

Therefore, each node of the quadtree can be represented by a quaternary code. After the quaternary code of the node is obtained, it can also be converted into the coordinates of the corresponding element in the pheromone matrix, where the ordinate is the quaternary code length of the node, and the abscissa needs to convert the quaternary code of the node into a decimal code.

### 4.2. Distance Matrix

The elements contained in the distance matrix are the path costs from each raster unit to other raster units, and the matrix dimension is $\left[4^{layer-1},4^{layer-1}\right]$. Ideally, the distance matrix should be a symmetric matrix, and the element value of the symmetry axis is 0. However, the path cost between raster units in a city may not be equal considering the differences in population flow and road planning, so the distance matrix may not be mathematically strictly symmetric. If the path cost between each raster units is difficult to calculate in two-dimensional space, the Euclidean distance or Manhattan distance between two raster centers can be calculated and stored in the distance matrix.

### 4.3. Ant Colony Location Algorithm

Taking the root node of the quadtree as the starting point of the ant colony, the ant colony localization algorithm is run to select the region. The moving sequence of ants in the quadtree is layer0, layer1 and layer m until reaching the leaf node of the quadtree, which is the location of the target region. Note that the layer m reached by an ant colony in an incomplete quadtree is not always equal to the total layers of the quadtree. Each ant must compute transition probabilities for all optional child nodes before moving to the next child node. Further, $p_{ij}^{k}\left(t\right)$ is the probability that the *k*th ant in the *t*th generation of ants selects different nodes, where *i* represents the current node, *j* represents the selectable node, and the transition probability calculation formula is shown in Formula (6).
(6)$$p_{ij}^{k}\left(t\right)=\begin{array}{c} \left\{\begin{array}{c} \frac{{\left[\tau_{j}\left(t\right)\right]}^{\alpha }{\left[\eta_{ij}\right]}^{\beta }}{\sum_{j=0}^{T_{i}^{k}\left(j\right)}{\left[\tau_{j}\left(t\right)\right]}^{\alpha }{\left[\eta_{ij}\right]}^{\beta }},if\;j\in T_{i}^{k}\\ 0,\;otherwise \end{array}\right. \end{array}$$
where $T_{i}^{k}\left(j\right)$ represents the set of optional child nodes contained in node *i* where the *k*th ant is located; $\tau_{j}\left(t\right)$ represents the pheromone of the optional transfer node *j*; $\eta_{ij}$ is obtained by the reciprocal of the path cost between nodes i∼j, in other words, $\frac{1}{d_{ij}}$; $d_{ij}$ is the path cost between i∼j nodes in the distance matrix; α is the pheromone heuristic factor; β is the expectation heuristic factor; and α and β are hyperparameters. After calculating the transition probabilities of all optional child nodes, use the roulette algorithm to select the next node. The individual selection probability of each optional node *j* in the roulette algorithm is shown in Formula (7): (7)$$q_{ij}=\frac{p_{ij}}{\sum_{j=0}^{T_{i}^{k}}p_{j}}$$
where $p_{ij}$ is calculated by Formula (6) to obtain the transition probability of the node.

After calculating the transition probability of each optional node, the individual probabilities of all nodes are accumulated to obtain the cumulative probability. The cumulative probability formula is shown in Formula (8): (8)$$Q_{ij}=\sum_{l=0}^{j}q_{il}$$

The cumulative probability transforms the probability space of the optional nodes into a linear interval, and generates a random number in the interval [0,1]. The interval where the random number is located is the selected node. Each ant moves from the root node until it reaches the leaf node of the quadtree according to the result of the roulette algorithm, which is the region selected by the ant colony localization algorithm.

### 4.4. Pheromone Update Method

Each generation of ant colony is based on the Ant-Cycle model and combines with pheromone diffusion to recursively update the pheromone matrix layer-by-layer according to the selected path. According to the movement path of the ant colony, the leaf nodes on the path of the ant colony are updated, and the pheromone diffusion method is used to update the pheromone of the neighborhood of the raster units represented by the leaf nodes in the raster space. The non-leaf nodes in the quadtree are equal to the sum of the pheromones of the child node contained after the update.

Using the pheromone diffusion algorithm in a nonuniform space is more complicated than in a uniform space. Therefore, the use of the pheromone diffusion algorithm in nonuniform spaces can be simplified:If the length and width of the raster unit in the target region selected by the ant are greater than the minimum raster unit, as shown in Formulas (4) and (5), then if the layer of the selected leaf node is not equal to the quadtree depth layer, only the pheromone of the region represented by the node is updated, and pheromone diffusion is not performed.If the length and width of the raster unit in the target region selected by the ant is equal to the minimum raster unit, but there are rasters whose length and width are greater than the minimum raster unit in its neighborhood, as shown in Formulas (4) and (5), it means that the layer of the leaf nodes represented by the neighborhood of the target region is not equal to the quadtree depth layer. Then, only the pheromone whose length and width are equal to the minimum raster unit in the neighborhood of the raster unit is updated.In Formula (6) using the ant colony localization algorithm to calculate the transition probability between nodes, $\tau_{j}\left(t\right)$ represents the pheromone of the *t*th generation of ants on the *j*th node of the quadtree. If node *j* is not a leaf node, the pheromone of all child nodes *l* included in node *j* is accumulated; the pheromone update formula for a non-leaf node is shown in Formula (9):
(9)$$\tau_{i}\left(t+1\right)=\sum_{T_{i}^{k}\left(j\right)}\tau_{j}\left(t\right)$$
where $T_{i}^{k}\left(j\right)$ is consistent with the above and represents the set of optional child nodes *j* included in node *i* where the *k*th ant is currently located. Based on Formula (9), the non-leaf node pheromones in the ant colony path can be updated layer-by-layer by a recursive method.If node *i* is a leaf node, the pheromone update formula of the leaf node of the path selected by each ant colony is shown in Formula (10). The neighborhood pheromone update formula of a leaf node is shown in (11):
(10)$$\tau_{i}\left(t+1\right)=\left(1-\rho \right)\tau_{i}\left(t\right)+▵\tau_{i}$$
(11)$$\tau_{i}^{Neighbor}\left(t+1\right)=c\cdot \left[\left(1-\rho \right)\tau_{i}\left(t\right)+▵\tau_{i}\right]+\tau_{i}^{Neighbor}\left(t\right)$$
where ρ is the pheromone polatility. Assuming that a total of *m* ants in the *t*th generation select node *i*, $▵\tau_{i}$ is the sum of the pheromone released by *m* ants on node *i*; $▵\tau_{i}$ can also be multiplied by the pheromone polatility, which is $\rho ▵\tau_{i}$; $\tau_{i}^{Neighbor}\left(t\right)$ represents the pheromone of the neighborhood raster in the raster space when the *t*th generation ant colony is located at leaf node *i*; *c* is the coefficient for updating the pheromone in the neighborhood of leaf node *i*, and the value range is [0,1]; the calculation formula of $▵\tau_{i}$ is shown in (12):
(12)$$▵\tau_{i}=\sum_{k=1}^{m}▵\tau_{i}^{k}$$
where *m* is the total number of ants, and the pheromone $▵\tau_{i}^{k}$ released by the *k*th ant on leaf node *i* can be calculated by Formula (13).
(13)$$▵\tau_{i}^{k}=\frac{Q}{L_{k}}$$
where *Q* is the pheromone constant, and the above coefficients ρ and *c* are both hyperparameters. Here, $L_{k}$ is improved on the concept of global update based on the Ant-Cycle model to represent the optimization objective between each positioning point. The calculation method of $L_{k}$ is introduced in detail in the fifth subsection.

### 4.5. Optimization Object

In the process of the ant colony localization algorithm for the location of the target region, after the ant colony movement is completed, the pheromone matrix needs to be updated according to the pheromone update formula. In the pheromone update (Formula (13)), it is necessary to obtain the optimization target $L_{k}$ between the raster units where the target region of the ant colony generation is located.

#### 4.5.1. Compactness Ratio

The compactness ratio is a spatial optimization objective based on the compact city theory. The concept of compactness ratio can accurately express the strength of the spatial gravitational force of the target region and can reflect the compactness of the spatial distribution of the target region. The core concept is high-density, efficient utilization of space resources and relatively centralized setting of target facilities. The compactness ratio calculation is shown in Formula (14): (14)$$L_{k}=\frac{\frac{N(N-1)}{2}}{\sum_{i=1}^{S}\sum_{{}_{j\neq i}^{j=1}}^{S}\frac{Z_{i}Z_{j}}{\mu d_{ij}^{2}}}$$
where *S* represents the number of independent raster units in the target region; $Z_{i}$ and $Z_{i}$ represent the area of raster unit *i* and raster unit *j* of the target region, respectively; $d_{ij}$ is the path cost between raster unit *i* and raster unit *j*, which can be obtained from the distance matrix; *N* is the total area of the two-dimensional space; and μ is a constant used to weigh the importance of the compactness ratio in different scenarios. Compactness encourages the aggregation effect when selecting the target region, and the distribution should be as regular and compact as possible, but it may be difficult to connect these target regions.

#### 4.5.2. Connectedness Ratio

Spatial continuity is an important spatial feature when selecting a target region and means that a region can reach any other point without leaving the space. For raster data, the degree of connectivity between such raster units can be measured by the connectedness ratio. The purpose is to connect the raster units in the target region in two-dimensional space. The connectedness ratio calculation is shown in Formula (15): (15)$$L_{k}=\frac{\left[\sum_{i=1}^{S}Z_{i}\right]\left[\sum_{i=1}^{S}Z_{i}-1\right]}{\sum_{i=1}^{S}Z_{i}\left(Z_{i}-1\right)+\sum_{i=1}^{S}\sum_{{}_{j\neq i}^{j=1}}^{S}\frac{Z_{i}Z_{j}}{\mu d_{ij}^{2}}}$$
where *S* represents the number of independent raster units in the target region; $Z_{i}$ and $Z_{i}$ represent the area of raster unit *i* and raster unit *j* of the target region, respectively; $d_{ij}$ is the path cost between raster unit *i* and raster unit *j*, which can be obtained from the distance matrix; *N* is the total area of the two-dimensional space; and μ is a constant used to weigh the importance of the connectedness ratio in different scenarios. The connectedness ratio encourages each raster unit to be connected with a smaller path cost when selecting the target region, which reflects the degree of dispersion between the target regions.

Both the compactness ratio and the connectedness ratio are spatial optimization targets for ant pheromone release in ant colony location algorithms. However, in the real world, other optimization objectives can be used depending on the application scenario. For example, economic optimization objectives based on gross domestic product (GDP), ecological optimization objectives based on compatibility, and a multi-objective optimization model that weighs parameters such as space, economy and ecology are used to optimize $L_{k}$ in the ant colony localization algorithm.

Repeat the above steps until the ant colony localization algorithm converges. The localization method of ant colony optimization in nonuniform space calculates the coverage of the target region and the weight of various influencing factors in the region in the initial two-dimensional space division. In the subsequent execution of the ant colony localization algorithm, the raster unit weights are used as the main basis to locate the target region. In the process of locating each ant colony, the path cost between each target region is considered and added into the pheromone updating formula as a parameter. Therefore, according to the above steps, the localization method of ant colony optimization in nonuniform space is the target region location result obtained by comprehensively considering the coverage, region weight and path cost.

## 5. Experiment

### 5.1. Dataset

The dataset used is city taxi data from ShangHai, where the longitude is about 120.83 to 121.98, and the latitude is about 30.66 to 31.87. Assuming that the entire research area can be divided into 64×64 units by the smallest raster, a 4−layer quadtree can be generated. The dataset comes from Didi’s taxi driving data in Shanghai, which includes the driving records of Didi’s vehicles at 18:00 on 11 July 2018. The key fields of each record include vehicle number, latitude and longitude, whether it is no-load, sampling time, speed and direction, etc. The dataset includes about 3,594,300 driving records. In this paper, the vehicle positions obtained at different sampling times in the dataset are aggregated into each raster unit. Figure 3a visualizes the number of vehicles contained in each unit in the form of a heat map. Figure 3b is the modeling results of the Shanghai city map for an incomplete quadtree, in which green is farmland and woodland, gray is mountains and dark blue is a lake. Since Didi has recorded these data for a long time, we assume that these driving records in the dataset are true, complete and reliable.

### 5.2. Settings

First, when the total amount of data is about 3,594,300 and it is used as the initial pheromone matrix, *Q* cannot be too small. If *Q* is too small, the concentration of pheromones released by the ant colony through the node is insufficient, and the weight of the node in the path cannot be effectively increased. According to the target function of location selection, multiple targets are selected for calculation to verify the effectiveness of the algorithm. Second, the study area is Shanghai, covering an area of 6340 km^2^. We divide Shanghai into four spatial scales according to the raster cell precision, which are 64×64, 128×128, 256×256 and 512×512, to observe the time cost of the ant colony algorithm in nonuniform space. Finally, we adjust the hyperparameters α, β, ρ and *c* in the algorithm to observe the ant colony diffusion and convergence by visualizing the pheromone concentration. The experiment uses a CPU with parameters of eight cores and 3.3 GHZ. The programming environment is python. Finally, this paper strictly ensures that subsequent comparative experiments are carried out under the same conditions.

### 5.3. Results

In this chapter, we observe the time cost of different ant colony numbers and iterations under four types of grid precision and compare the running effect of the classical ant colony algorithm and the ant colony algorithm proposed in this paper in nonuniform space. The parameters (α,β,ρ,c) in the algorith equal (0.5,1,0.3,0.3), respectively. Algorithm efficiency comparisons are often made based on computer CPU time. The overall time span of the ant colony algorithm in nonuniform space is between 1–300 s, while the longest time for the classical ant colony algorithm is 1421 s. It can be seen from the results that the ant colony algorithm combined with the incomplete quadtree can greatly reduce the time cost under the same conditions. In the initial state, the time cost does not change significantly, but with an increase in the number of iterations, the number of ant colonies and/or the raster accuracy, the running time obviously increases. The raster cell accuracy has the greatest influence on the algorithm time cost, followed by the number of iterations, and the ant colony number has the least influence on the algorithm time cost. The Table 1 is the classical ant colony algorithm and the Table 2 is the Ant colony optimization in nonuniform space.

Figure 4 shows the convergence of ant colonies of different scales, taking Shanghai as an example. After several iterations, the ant colonies finally converge on the raster with the highest pheromone concentration. Depending on the size of the ant colony, the site selection of the raster changes as shown below.

It can be seen from Figure 4 that the site selection results meet the requirements of the problem setting. When the number of sites is two, the results of site selection are mainly concentrated in the densely populated Huangpu District and Jing’an District. With the increase in the number of site selection targets, site selection spreads from Huangpu District and Jing’an District to Changning District, Hongkou District, Xuhui District and Pudong New District. When the target location exceeds 10, the location target expands to the entire space. The overall variation law of site selection in space is as follows: the sites selected in densely populated central urban areas account for a larger proportion. The peripheral areas with low population density, generally along the central urban area to Qingpu District (to the west), from the central area to Pudong New Area (to the east), from the central area to Jiading District (to the northwest), and from the central area to Minhang District (to the south) have incremental expansion with site selections in several axis directions. With the continuous increase in the number of sites, the specific locations of sites is more and more consistent with the location of the population.

In real life, incomplete quadtree modeling is suitable for space situations that cannot be fully utilized. Under the condition of ensuring resolution, the location results of the ant colony algorithm are consistent with the direct calculation results, but the incomplete quadtree modeling takes less time to calculate. However, the modeling process of incomplete quadtrees is more complicated than that of direct spatial meshing because the modeling process of incomplete quadtrees requires users to have deeper understanding and research of the local situation. Local conditions include the urban spatial structure and the flow of people. Furthermore, the setting of the number of ants in the algorithm is usually done empirically. When the space is large, more ants can improve the global searchability and stability of the algorithm, but too many ants reduce the solvability of the location model. If the number of ants is too small, the randomness of ant search is weakened, convergence is accelerated, it is easy to fall into a local optimum and the phenomenon of premature maturity appears.

The advantage of the algorithm described in this paper is that the ant colony algorithm is used to combine the probabilistic selection strategy. When the ants in the ant colony algorithm choose a path, they choose the current best path with a certain probability. Even if the initial distribution of ants corresponds to a better solution or when it falls into a local extreme point at a certain moment, the entire ant colony continues to approach the global optimal solution. Second, the quadtree structure in spatial modeling searches the geographic space. Due to the logarithmic relationship between the search scale and the space scale, the quadtree structure in spatial modeling shows a slow growth trend with the increase of the space scale. This approach allows the algorithm to maintain a fast search speed and is suitable for solving location problems in large-scale spaces. Finally, meta-heuristics need to strike a balance between optimality and efficiency in order for an approach to perform well. However, there is no single approach that can fully achieve this balance. This balancing problem is a hyper-optimization problem that depends on many factors, such as how the method works, parameter settings and control over parameters. Thus, this balance may not be universal. The main means to achieve such a balance in this paper are to control the modeling accuracy of the incomplete quadtree and to implement the two ant colony optimization objectives presented in this paper. Of course, the goal of ant colony optimization is not only compactness and connectivity, and users can expand it according to the actual situation.

## 6. Conclusions

Under the constraints of spatial scale and spatial precision, it is often difficult to obtain satisfactory solutions in terms of space and time efficiency by conventional algorithms for multi-objective spatial location problems. The heuristic algorithm has become an effective tool to solve the problem of multi-objective spatial location because of its strong robustness, excellent distributed computing mechanism, and easy combination of data structure with other methods. The localization method of ant colony optimization in nonuniform space proposed in this paper uses the strategy of multi-level nonuniform division in space. The ant colony moves on the nodes of the incomplete quadtree and communicates and cooperates indirectly through pheromone so as to find high-quality candidate solutions. During the search process of the ant colony, the quadtree needs to be initialized based on social factors (economy, population, etc.). According to the pheromone concentration, the current best path is selected with a certain probability so as to ensure that the ants search along the best solution path, but we also use the pheromone concentration diffusion algorithm to improve the probability of selecting other paths during the movement process to avoid getting trapped in local extrema during site selection. From the moving process of the entire ant colony in the nonuniform space, the ant colony continues to approach the global optimal solution. In the experiment, traffic flow data in Shanghai are used to initialize the quadtree, and the ant colony location algorithm based on the multi-tree can find the ideal solution quickly and accurately. In the course of the experiment, we also found that the time cost of the algorithm increases proportionally with the increase of the space scale. The solution of the objective function and the region in the ant colony algorithm still has the problem of intensive computation. However, the quadtree data structure generated by nonuniform space and heuristic search algorithms such as the ant colony algorithm are suitable for distributed computing framework. Therefore, in follow-up research, parallel computing can be used to improve the multi-location algorithm of multi-ant colony intelligence in nonuniform space, aiming to improve the time efficiency of the algorithm in large-scale space.

## Figures and Tables

**Figure 1 sensors-22-07389-f001:**
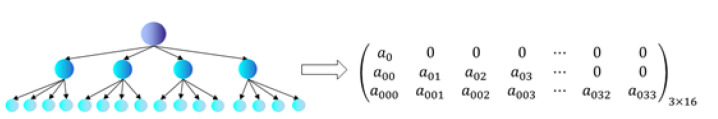
The process of initializing the pheromone matrix of the three-layer quadtree.

**Figure 2 sensors-22-07389-f002:**
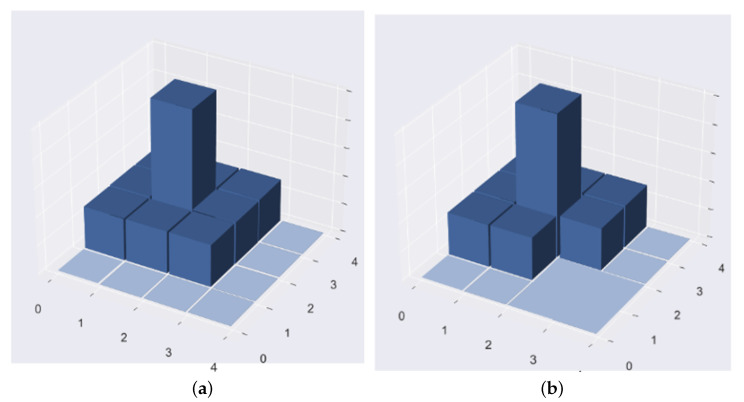
(**a**) Complete pheromone diffusion. (**b**) Incomplete pheromone diffusion.

**Figure 3 sensors-22-07389-f003:**
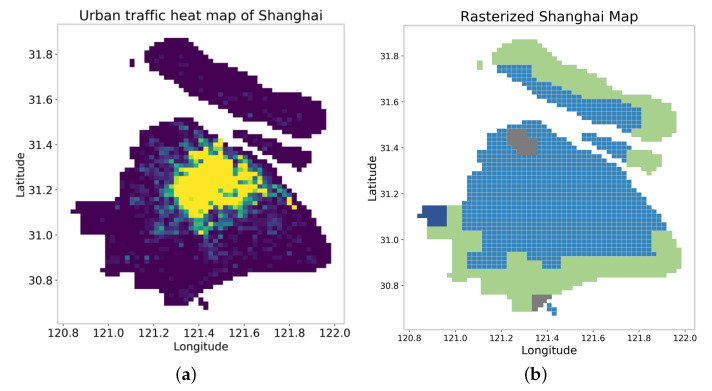
(**a**) Urban traffic heat map of Shanghai. (**b**) Rasterized Shanghai map.

**Figure 4 sensors-22-07389-f004:**
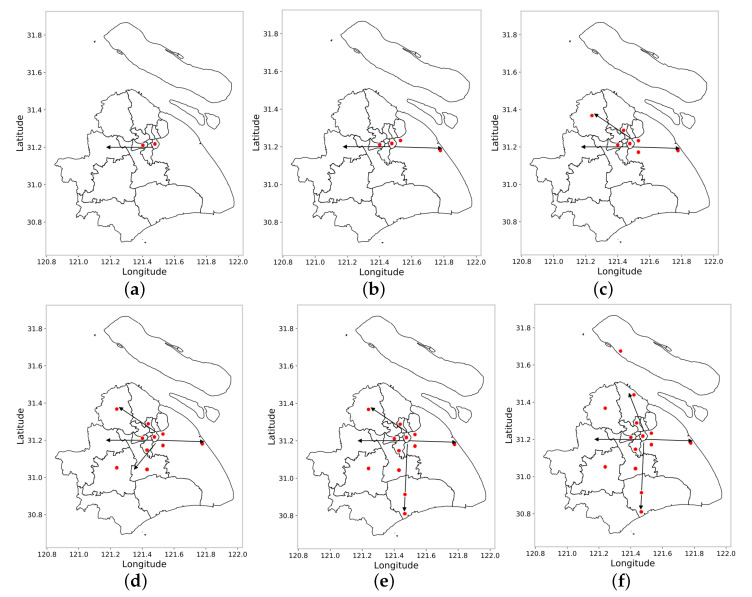
Site selection targets of (**a**) 2, (**b**) 4, (**c**) 7, (**d**) 10, (**e**) 12 and (**f**) 14.

**Table 1 sensors-22-07389-t001:** The classical ant colony algorithm.

Epochs		Ant Quantity	10	20	50	100
	Spatial Scale					
10	64 × 64		1.65 s	1.63 s	2.89 s	6.35 s
	128 × 128		7.73 s	16.84 s	25.52 s	66.3 s
	256 × 256		40.64 s	67.90 s	143.13 s	189.81 s
	512 × 512		124.45 s	187.79 s	245.93 s	432.80 s
50	64 × 64		3.81 s	4.10 s	7.74 s	12.22 s
	128 × 128		10.30 s	20.57 s	35.60 s	69.00 s
	256 × 256		49.40 s	87.36 s	160.89 s	303.11 s
	512 × 512		150.36 s	297.01 s	450.49 s	780.98 s
100	64 × 64		5.92 s	10.63 s	17.30 s	29.12 s
	128 × 128		8.61 s	33.44 s	54.82 s	121.50 s
	256 × 256		70.51 s	160.55 s	341.81 s	741.63 s
	512 × 512		263.63 s	460.80 s	803.08 s	1421.8 s

**Table 2 sensors-22-07389-t002:** Ant colony optimization in nonuniform space.

Epochs		Ant Quantity	10	20	50	100
	Spatial Scale					
10	64 × 64		1.48 s	1.77 s	3.19 s	5.05 s
	128 × 128		4.93 s	5.99 s	10.32 s	13.11 s
	256 × 256		10.33 s	18.64 s	35.03 s	50.39 s
	512 × 512		44.70 s	60.20 s	81.01 s	119.30 s
50	64 × 64		1.56 s	2.46 s	4.19 s	7.90 s
	128 × 128		5.10 s	10.79 s	13.11 s	20.55 s
	256 × 256		13.66 s	23.06 s	50.09 s	63.81 s
	512 × 512		50.19 s	70.07 s	120.70 s	180.22 s
100	64 × 64		1.92 s	2.63 s	5.13 s	9.52 s
	128 × 128		7.03 s	10.40 s	16.13 s	24.49 s
	256 × 256		20.51 s	32.60 s	62.00 s	83.84 s
	512 × 512		68.40 s	103.91 s	212.47 s	301.29 s

## Data Availability

Not applicable.

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
