# Peer review of "A Localization Method of Ant Colony Optimization in Nonuniform Space"

_sensors, 2022, doi:10.3390/s22197389_

Round 1
Reviewer 1 Report
The paper seems original and interesting results are presented. My concerns, questions, and suggestions are listed below:
1. The abstract should be more straightforward for the reader regarding the proposed method and its motivation. The abstract should present some main points for the readers, such as the main contributions, the proposed method, the main problem, the obtained results, the benchmark tests or the data, the comparative methods, etc.
2. The authors just described the related works that the researchers have done, but they did not evaluate the advantages and disadvantages of the related works. Please evaluate that how their study is different from others in the related work section? What do they have where others do not? Why they are better or how?
3. Motivation does not exist.
4. Organization of the paper is not presented at the end of Section 1.
5. Analysis of the results is missing in the paper. There is a big gap between the results and conclusion. There should be the result analysis between these two sections. After comparing the methods, you have to be able to analyze the results and relate them to the structure of all algorithms. It would be interesting to have your thoughts on why the method works that way? Such analyses would be the core of your work where you prove your understanding of the reason behind the results.
6. Some mathematical notations are not rigorous enough to correctly understand the contents of the paper. The authors are requested to recheck all the definition of variables and further clarify these equations.
7. The authors should discuss the literature review more deeply and clearly. Try to make the paragraphs in the introduction section more comprehensive, it is short. The current introduction and related works seem simple and misses some contents related to the problem formulation. There is not a clear categorization on the metaheuristic evolutionary methods and description of these algorithms are simplistic for this topic. These methods are categorized into nine different classes according to the papers entitled “Plant intelligence based metaheuristic optimization algorithms”, “Comparative Assessment of Light-Based Intelligent Search and Optimization Algorithms”, “A Physics Based Novel Approach for Travelling Tournament Problem: Optics Inspired Optimization”, and “Chaos Based Optics Inspired Optimization Algorithms as Global Solution Search Approach”. These papers should be cited for brief categorization of metaheuristic algorithms. In this current form there is confusion.
8. Clarifying the study’s limitations allows the readers to better understand under which conditions the results should be interpreted. A clear description of limitations of a study also shows that the researcher has a holistic understanding of his/her study. However, the authors fail to demonstrate this in their paper. The authors should clarify the pros and cons of the methods. The discussion of the results needs to include the strengths and weaknesses of the proposed algorithm. What are the limitation(s) methodology(ies) adopted in this work? Please indicate practical advantages, and discuss research limitations. These limitations can be organized around simple distinctions of the choices you made in your study regarding who, what, where, when, why, and how. To have an unbiased view in the paper, there should be some discussions on the limitations of the proposed method.
9. What are the other possible methodologies that can be used to achieve your objective in relation to this work?
10. Add further details on how simulations were conducted. Similarly, system and resource characteristics could be added to tables for clarity. The paper lacks the running environment, including software and hardware. The analysis and configurations of experiments should be presented in detail for reproducibility. It is convenient for other researchers to redo your experiments and this makes your work easy acceptance.
11. Are the simulation results taken from the equal conditions? There is not any discussion.
Reviewer 2 Report
There may be an interesting paper in this: I find the combination of ant algorithms and quadtrees innovative and the results presented, albeit preliminary and insufficiently detailed in my opinion, support the idea that the method has potential. Unfortunately, as it stands, the presentation makes this impossible to assess objectively: the level of English is very poor, especially the syntax (incomplete sentences etc.). I understand that English is not your first language (it isn't mine either), but it is necessary to maintain sufficient language standards for authors from different backgrounds to be able to understand and collaborate with each other. In its present state, your manuscript does not meet this requirement.
I am unable to fully assess the scientific quality due to the poor presentation, but here are some comments which you may find useful:
1. The first few sentences of the introduction are misleading. Although it is important to position a publication into the wider context, this is going much too far: it suggests to the reader that your paper is relevant to the study of epidemic diseases, which really isn't the case.
2. I cannot find any meaningful comparison with other methods to solve similar problems. This would not necessarily be an issue if the topic was unusual, but the optimal positioning of resources in a heterogeneous environment is an optimisation classic. Furthermore, you make some claims about your solution being efficient and scalable, which I do not dispute, but compared to what?
3. While on the topic of performance, I would suggest compacting Table 1: the reader doesn't need to know the convergence time for every tested combination of parameter values (minor correction).
4. It is not clear to me how the quadtree is constructed. It seems clear that the main advantage of the method is that the "resolution" can vary from one region of the map to the other (i.e. leaves are at different depths). Some elements (e.g. Fig 2) confirm this, but others are confusing: for instance, Fig. 3 ("rasterized Shanghai heat map") looks like a homogeneous 64x64 regular lattice. I realise that these figures are just produced from the urban traffic data that was used as a proxy of population density(?), but this is precisely my point. Aside from the fact that Fig. 3 left and right are somewhat redundant, it would be much more useful and interesting to see a visualisation of the heterogeneous quadtree version of the map (or region thereof if scale is an issue) in which the ant colonies operate.
5. As far as I can tell, Fig. 4 is the only place where the output of your numerical experiments is actually presented, in six tiny maps. This is a major problem: you go to great lengths to describe what you did (although the poor presentation means it's difficult to follow and would be impossible to reproduce with the information provided), but almost gloss over the actual results. This needs complete re-balancing, meaning potentially a (much) longer paper.
In summary, the main issue is the presentation, not the content. I think you have enough material to publish, but it just isn't there, at least not in a meaningful and usable format. Please put some effort into making your work understandable by others. At present, the manuscript is just sloppy. Even the figure captions are botched up: fig. 2, 3 and 4 have the same! Sadly, this is just an example of the poor presentation in general...
Regarding language, I would suggest seeking help from professional editing/translation services.
Round 2
Reviewer 2 Report
I appreciate that you have done some corrections. I am not convinced they are sufficient but will defer to the editor's decision.